# Temporal Chain of Thought:
# Long-Video Understanding by Thinking in Frames

**Anurag Arnab**[*]   **Ahmet Iscen**[*]   **Mathilde Caron**   **Alireza Fathi**   **Cordelia Schmid**
Google DeepMind

## Abstract

Despite recent advances in Vision-Language Models (VLMs), long-video understanding remains a challenging problem. Although state-of-the-art long-context VLMs can process around 1000 input frames, they still struggle to effectively leverage this sequence length, and succumb to irrelevant distractors within the context window. We present Temporal Chain of Thought, an inference strategy for video question-answering that curates the model's input context. We use the VLM itself to iteratively identify and extract the most relevant frames from the video, which are then used for answering. We demonstrate how leveraging more computation at inference-time to select the most relevant context leads to improvements in accuracy, in agreement with recent work on inference-time scaling of LLMs. Moreover, we achieve state-of-the-art results on 4 diverse video question-answering datasets, showing consistent improvements with 3 different VLMs. In particular, our method shines on longer videos which would not otherwise fit within the model's context window: On longer videos of more than 1 hour on LVBench, our approach using a context window of 32K outperforms the same VLM using standard inference with a 700K context window by 2.8 points.

## 1   Introduction

Despite recent advances in Vision-Language Models (VLMs) [6, 31, 32, 34, 55], understanding long videos remains a challenging problem. This difficulty stems from the fact that this task requires a VLM to process a long sequence of input tokens, and requires the model to possess a host of interrelated abilities including action and scene understanding, long-term memory, and tracking state changes and interactions among others. The long-context ability of leading VLMs, that enables models to process hundreds or even a thousand frames of input context [36, 55, 56], is a valuable step forward in this regard. However, numerous studies have shown that processing longer contexts can saturate or degrade accuracy, as the model is overwhelmed with irrelevant or misleading content [23, 28, 37, 64].

Based on the observation that too large of an input context can be distracting, we propose an inference strategy, Temporal Chain of Thought, which first aggregates relevant context from the input video, and then uses it to answer the question (Fig. 1). Prior works based on the principle of removing distracting context from a video [28, 45, 61, 62] used an ensemble of multiple models, typically using one model to caption individual frames, another to find relevant ones, and finally answer the question with an LLM. In contrast, we use only a single VLM to both select the relevant context, and to answer the question, and show how our inference strategy provides substantial improvements.

Our approach is motivated by recent studies in Large Language Models (LLMs) which suggest that scaling inference-time computation is more effective than scaling the number of model parameters [9, 20, 52, 65]. Similarly, we show how leveraging more computation to aggregate relevant information from the video results in higher accuracy. In addition, as our approach iteratively extracts relevant

---

[*]Equal contribution.

39th Conference on Neural Information Processing Systems (NeurIPS 2025).

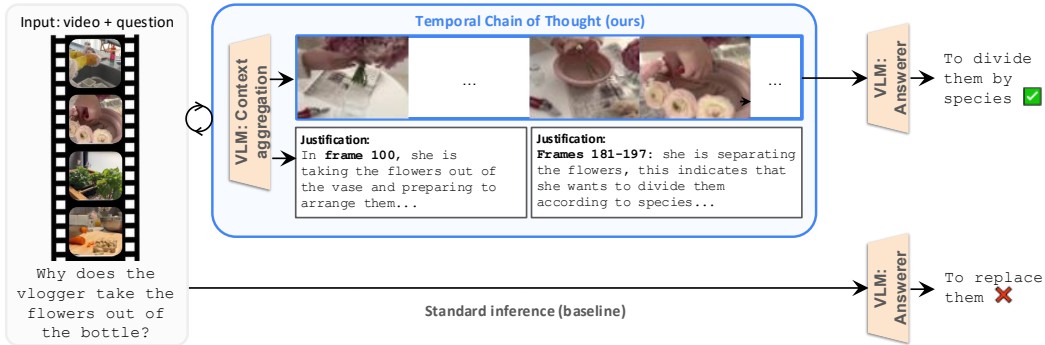

Figure 1: **Temporal Chain of Thought.** Motivated by the fact that long input contexts can have distractors which confuse the model, we use the VLM itself to first extract relevant context (blue box) before processing it. Our approach improves accuracy, and by iteratively processing parts of the video at a time, can also reduce the model's required context window.

context from the video, it means that we can effectively process videos that would not otherwise fit within the model's context limit. Moreover, our approach has connections to Chain-of-Thought prompting [63] (and multi-step extensions of it [59, 68]) in language, where the model is prompted to first output textual "thoughts" which help it to produce the final answer. As we aggregate the relevant frames in the video, we can think of these frames as "visual thoughts". Furthermore, as a by-product, we can use the model's justifications for choosing relevant frames to interpret it (Fig. 1).

We find that the general principle of aggregating relevant context from a video is beneficial for video question-answering (QA), with our proposed method consistently improving results across 4 datasets and 3 different VLMs. For shorter videos, on the order of hundreds of frames, our inference strategy improves results even when the entire video could fit within the model's context window, emphasising that by removing distractors from the input, we can improve the model's reasoning ability.

For longer videos, on the order of a thousand frames, such as LVBench [58], our method shows even larger improvements. Given a fixed context-window budget for a VLM, our model can iteratively extract the most relevant context leading to substantial accuracy gains. Furthermore, as our inference strategy focuses the model on the most relevant context, we can even outperform standard, baseline inference with a much longer context window.

In summary, we propose the following contributions:

- A novel VLM inference strategy for video QA.
- Thorough experimental analyses confirming that the principle of context aggregation is effective, that our approach outperforms standard long-context inference across a range of computational budgets, is adaptive to the question type and generalises to multiple VLMs.
- State-of-the-art results on 4 video understanding benchmarks. In particular, on LVBench where videos average 68 minutes in length, we improve by 11.4 points given a context-window budget of 32K tokens. Moreover, our iterative approach using a 32K context-limit outperforms a long-context baseline using the same 700K total tokens by 2.8 points.

## 2 Related Work

**Long-context LLMs**   Our work is motivated by several prior studies, predominantly in the domain of natural language, which have demonstrated how Large Language Models (LLMs) are not able to effectively leverage their full input contexts [23, 37, 64, 67, 70, 74]. By asking questions where the position of a crucial piece of information in the input context is varied in a controlled manner, studies have shown that performance degrades considerably when the relevant context is not at the beginning or end of the input sequence [37, 64, 70].

**Long-context video understanding**   The most related works for long-video understanding, however, are [5, 21, 28, 45, 57, 61, 62, 72]. These works represent a long video by first computing captions at each frame of the video with a dataset-specific model [22, 35, 73], which are then fed to an LLM along with the input question to produce an answer. Observing that redundant captions degrade the performance of the "answerer" LLM, a number of strategies have been proposed: Video Agent [61] begins from uniformly sampled frames, and uses EVA-CLIP embeddings [53] to iteratively retrieve frames until sufficient context has been obtained for GPT-4 to output an answer it is confident in. Video Tree [62] in contrast clusters per-frame captions together (where each cluster is then represented

by the caption of the frame closest to the centroid), and then only uses the top-scoring centroids in the final answering phase. Language Repository [28] aggregates per-frame captions into a global textual representation of the video, using an LLM to summarise different captions together, and CLIP similarities to prune redundant captions. VideoRAG [39], in contrast, supplements the video with auxilliary model outputs [24, 27, 46, 50], namely object detection, ASR and OCR, represented as text. Similarly, [16, 42, 54] call external tools based on per-frame captions to answer questions.

Although we also extract relevant context from the input video, our approach is fundamentally different in that we operate directly on video frames, and not captions as an intermediate representation. As we do not rely on initial per-frame captions, our approach is not limited by the captioner missing details relevant to the question (particularly because the captioning in these works is not conditioned on the input question). Moreover, we use only a single VLM in the entire inference process unlike the aforementioned works, which means that our approach is conceptually more elegant and simpler to deploy. The fact that we use a single model to generate intermediate outputs (the relevant frame indices to the question) is akin to "Chain-of-Thought" [29, 63] prompting.

**Chain-of-Thought** Chain-of-Thought (CoT) [63] was originally developed for few-shot prompting of LLMs for symbolic reasoning or arithmetic tasks. Concretely, the model is prompted to first output (in natural language) the steps first required to solve a reasoning problem, before outputting the final answer. This approach is effective as the LLM is conditioned on the initial reasoning (or "thoughts") before making its final prediction during autoregressive decoding. Subsequent works extended this approach, proposing inference-scaling strategies involving multiple LLM calls: The LLM first produces multiple hypotheses, which are then ranked, and the top-scoring ones explored further [9, 43, 51, 52, 59, 68]. Our method has similarities to [63], but instead of producing "thoughts" as language, predicts relevant frame indices in the video instead. Note that for video, Fei *et al.* [17] first predict spatio-temporal scene graphs [26] and use these intermediate representations for answering the question. Our method is more general, as it is not object-centric like [17], and our iterative approach for finding the most relevant frames has analogues of [52, 59, 68] for video understanding.

**Token- or frame-selection** Finally, numerous prior works learn neural network layers to reduce the number of tokens that need to be processed by a subsequent transformer [8, 14, 25, 48, 67, 75]. Among these, models specialised for video typically learn to select individual frames [10, 11, 30, 33, 49, 60]. However, these methods need to backpropagate gradients through the subsequent transformer during training, meaning that it is not computationally feasible to employ such approaches with the largest, pretrained VLMs [3, 55] as our work. Such models are also trained for a fixed, small number of input frames (*i.e.* 32 for SeViLA [69] and ViLA [60]), whilst we show that our approach can handle thousands. Moreover, perhaps surprisingly, we demonstrate that we do not need separate networks to select relevant frames, but can use the VLM itself to do so in an effective and adaptive manner.

## 3 Proposed Approach

We begin by reviewing the standard inference process for Vision Language Models (VLMs) before describing our proposed Temporal Chain of Thought.

### 3.1 Standard VLM inference

The standard inference approach for VLMs to answer a question, $\mathbf{q}$, about an input video, $\mathbf{x} \in \mathbb{R}^{T \times H \times W \times C}$, is to simply forward it through the model, $f$,

$$\mathbf{a} = f(\mathbf{x}, \mathbf{q}), \tag{1}$$

where $f$ denotes the VLM, $T$, $H$ and $W$ denote the temporal- and spatial dimensions respectively, and $\mathbf{a}$ the predicted answer, where $\mathbf{q}$ and $\mathbf{a}$ are sequences of language tokens which index a discrete vocabulary, $\mathcal{V}$. Note that visual inputs, $\mathbf{x}$, are typically projected, or tokenised into the same space as language tokens. As the overall sequence length of the model is limited by computation, the frames of the video typically need to be subsampled to fit within the context-limit, $k$, of the model. Current models with the longest context windows can typically fit videos of up to one hour at 1 fps [36, 55].

### 3.2 Temporal Chain of Thought

Our method is motivated by the fact that although VLMs are now capable of handling increasing large input context lengths [36, 55, 56] they often still struggle to leverage this effectively and are confused by irrelevant distractors within this large context [23, 28, 37].

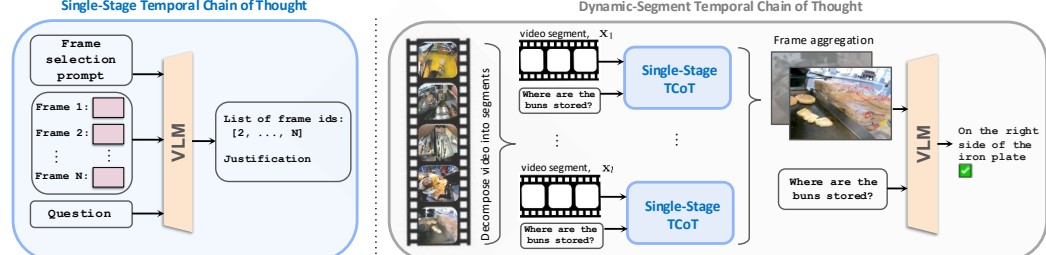

Figure 2: **Temporal Chain of Thought**. We use Single-Step TCoT (left, Sec. 3.2) to construct our final approach (right). Namely, we use the VLM itself to extract relevant frames from an input video clip, conditioned on the input question. To scalably process longer videos, we perform this approach within $l$ segments which span the video to extract the most relevant context. Finally, we use only the extracted context for answering.

Therefore to answer a question, $\mathbf{q}$, about an input video, $\mathbf{x}$, we do not directly pass both inputs to the model directly. Instead, we decompose video question-answering into first extracting the relevant context, $\mathbf{c}$, from the input $\mathbf{x}$, and then answering using $\mathbf{c}$ instead (Fig. 1). Crucially, this decomposition is performed by the same instruction-tuned VLM which will perform the subsequent answering, drawing inspiration from Chain-of-Thought and related LLM inference strategies [52, 63, 68].

Formally, our inference procedure consists of two stages: First we assemble the relevant context, $\mathbf{c}$, from the input video $\mathbf{x}$ and question $\mathbf{q}$. Thereafter, we answer the question. We denote these stages as

$$\mathbf{c} = G(\mathbf{x}, \mathbf{q}), \tag{2}$$
$$\mathbf{a} = H(\mathbf{c}, \mathbf{q}), \tag{3}$$

where we denote $G$ as the context aggregation function, and $H = f(\mathbf{c}, \mathbf{q})$ is the answering function, which simply forwards the extracted context through the VLM.

Note that $G$ itself can be a multi-step inference process. It is also adaptive in that the number of context tokens, $\mathbf{c}$, that are selected is input-dependent as long as it fits within the VLM's context limit, $k$. Importantly, we use the same VLM for both $G$ and $H$ and do not rely on any external models or tools. Note that by carefully designing $G$, we can also effectively process long videos which would otherwise not fit in the model's context limit.

We first introduce a simple form of $G$ next, Single-Step Temporal Chain of Thought (TCoT), which we use to construct our final approach, Dynamic-Segment TCoT.

**Single-Step TCoT** In this simple approach (Fig. 2, left), which is the basis of our final method, we simply query the VLM for what frames it needs to answer a question. Given up to $N$ frames from the input video, $\mathbf{x} = [x_1, \ldots, x_N]$, which fit the context-limit of the model, we prompt the model to output the frame ids which are relevant to answering the given question as

$$\hat{\mathbf{x}}, \mathcal{S}, \mathbf{j} = S(\mathbf{x}, \mathbf{q}), \tag{4}$$

where $S$ denotes the VLM selection call (prompt in Fig. 3), $\mathcal{S}$ the selected frames, and $\mathbf{j}$ a textual justification of this decision, which can be used for interpreting the model's predictions (Fig. 1). $\hat{\mathbf{x}}$ is the resampling of the input $\mathbf{x}$ using the frame ids in $\mathcal{S}$, namely $\hat{\mathbf{x}} = \{x_i, \ldots, x_j\}$ for $i, \ldots, j \in \mathcal{S}$.

We validate $\mathcal{S}$ to ensure that the frame indices are in ascending order, contain no duplicates and are within bounds. For simple parsing, we prompt the model to predict in the JSON format [15], and for responses that fail to parse, we assume that $\mathcal{S} = [1, \ldots, N]$ is all frames within the video.

```
You will be given a question about a video and five
possible answer options.

FrameID 1:{frame1},...,FrameID N:{frame N}
Question: {question}
Possible answer choices: {answer choices}

Return the frame ids which can answer the given
question.

Please use the following JSON format for your
output:
{'frame_ids': [List of integer frame IDs],
 'justification': Justification about your output}
```

Figure 3: **Prompt for our VLM selection call**, $S$, (Eq. 4).

Additionally, in practice, we found that the relevant frames, $\hat{\mathbf{x}}$ are sometimes too succinct, which may make it more difficult for the answerer, $H$, to answer from $\hat{\mathbf{x}}$ alone. For example, as shown in App. C, for the question, *On what floor is the washing machine?*, the VLM correctly localises the washing machine, but it is otherwise difficult to discern which floor of the house it is on. To remedy this issue, we also include a small number, $u$, of uniformly sampled frames from the original input, $\mathbf{x}$, denoted as $\mathbf{x}_{[u]}$, where $u \ll N$. Thus, the final context aggregated is $\mathbf{c} = \hat{\mathbf{x}} \cup \mathbf{x}_{[u]}$.

This simple approach removes redundancy in the input $\mathbf{x}$ which may otherwise confuse the model [10, 28, 37, 69]. However, it does not scale to long input sequences. This is because we need to simultaneously process all frames, which may exceed the model's context limits, and this subtask of finding relevant frames in a long video is itself prone to distractors. We address these issues next.

**Dynamic-Segment TCoT**    To decouple the video length from the context limit of the model, and to overcome limitations in frame recall from uniformly sampling input frames to fit within the model's context limit, we design Dynamic-Segment TCoT. As shown in Fig. 2, we partition the videos into $l$ separate segments, which we process independently, before aggregating them to form a holistic video representation.

To process the input video $\mathbf{x}$, comprising $N$ frames, we divide it into $l$ non-overlapping segments of equal length. We denote each segment as $\mathbf{x}_i = \{x_{(i-1)m+1}, \ldots, x_{im}\}$, where $i \in \{1, \ldots, l\}$ indexes the segment, and $m = N/l$ represents the number of frames per segment.

Note that we may have a large segment size, $m$ when either the video is long (large $N$) or few segments (small $l$). This presents a challenge as firstly, the segment length may exceed the model's context limit, $k$. Second, larger segments may contain more distractors, making it harder to identify relevant frames. To mitigate these issues, we uniformly sample $s$ frames from each segment $\mathbf{x}_i$, denoted as $\mathbf{x}_{i_{[s]}}$ This ensures a tractable number of frames are considered, balancing computational cost with temporal coverage. The sampled frames from each segment are then processed independently, and the resulting outputs are concatenated to form the final representation:

$$\hat{\mathbf{x}} = [S(\mathbf{x}_{1_{[s]}}, \mathbf{q}), \ldots, S(\mathbf{x}_{l_{[s]}}, \mathbf{q})]. \tag{5}$$

As we processed segments independently, it is possible that $\hat{\mathbf{x}}$ is larger than our context-window limit, $k$. Moreover, we also find it beneficial to include coarse, uniform context, $\mathbf{x}_{[u]}$ as before. Therefore, if $\hat{\mathbf{x}}$ contains more frames than $k$, we refine $\hat{\mathbf{x}}$ by uniformly sampling $m = k - u$ frames from it, and adding $u$ uniform context frames. Therefore, the final context aggregated is $\mathbf{c} = \hat{\mathbf{x}}_{[m]} \cup \mathbf{x}_{[u]}$.

**Discussion**    Partitioning a video into $l$ segments ensures that we can process a long-video with a fixed computational cost, regardless of the video-length, $N$. For standard VLM inference, the cost grows with the length of the video, and is limited by the maximum supported context-limit, $k$. In contrast, the required context-length with our approach is always fixed at $s$, to process a total of $s \cdot l$ frames. Note that our method not only decomposes a video into smaller segments, but importantly reasons across them in the answering stage, as analysed in Tab. 4 of App. A.

Moreover, by varying the number of segments, $l$, we can smoothly increase both inference-computation and accuracy (which we show experimentally in the next section). These trends are in agreement with recent work in language [9, 52, 59] which also uses additional compute at inference time to solve challenging problems with LLMs.

## 4 Experiments

### 4.1 Experimental Setup

**Models**    We use Gemini 1.5 Flash as our primary VLM, specifically the *Gemini-1.5-flash-002* checkpoint via the Vertex API [18]. This is because Gemini is already the state-of-the-art on a number of video question-answering (QA) datasets (Tab. 3), and it supports long context lengths for fair comparison of baseline inference to our method. Gemini uses 258 tokens per frame, and unless otherwise specified, we use a context budget of 32K tokens. This corresponds to 120 frames leaving sufficient remaining tokens for the input question. To show the generalisability of our method, we also use Qwen-2.5-VL [6] and GPT-4o-mini [1]. The prompts for all models are detailed in App. D. For all datasets, we sample videos at 1 frame per second (fps) following [55, 58, 61, 72].

**Datasets**    We use the following long-video QA datasets:

*Egoschema* [41] is a popular benchmark derived from Ego4D [19]. It consists of 5-way multiple choice questions on videos which are 180 seconds long. We run ablations on the subset of 500 labelled examples, and also report results on the full set of 5000 videos via the evaluation server.

*LVBench* [58] is a recent dataset with an average length of 4080 seconds (68 minutes), and four multiple choice options. Videos are from YouTube, and since some are no longer available online, we include our full list of video ids in App. D. We use visual inputs only.

*OpenEQA* [40] is a recent dataset targeted at embodied QA for mobile agents. It adds open-ended questions to the HM3D [47] and ScanNet [13] datasets, where videos are on average 452 frames long. As the answers are open-ended, the standard protocol is to use an LLM, specifically GPT-4 [2], to score the predicted answer against the ground truth answer using a scale from 1 to 5. These results are then averaged and normalised to a score out of 100.

*NExT-QA* [66] is a popular video QA dataset with five multiple choice options. Videos are on average only 39.5 seconds long. However, we report results on this dataset to compare to prior works.

## 4.2 Ablation Studies

**Context Aggregation Analysis**    Table 1 compares different context aggregation functions, $G$, (Sec. 3.2) on both Egoschema and LVBench. Baseline inference performs no context aggregation and answers the question directly. We use a 32K token context limit, corresponding to 120 frames.

Table 1: **Comparison of different context aggregation approaches**. All methods take 120 frames as input, using a 32K context-window for Gemini Flash. Our improvements are larger on the longer LVBench dataset.

|  | Egoschema | LVBench |
|---|---|---|
| Baseline inference | 72.6 | 50.3 |
| Single-step | 74.8 | 48.3 |
| Hierarchical | 74.0 | 53.3 |
| Dynamic-segment | **75.2** | **61.7** |

In addition to Single-Step and Dynamic-Segment  TCoT (Sec. 3.2), we consider an additional approach which can iteratively process a long video, which we denote Hierarchical  TCoT. As detailed in App. D, we perform Single-Step  TCoT iteratively: First we coarsely sample frames from the video, then once we have identified frames of interest, we sample nearby frames that were not initially considered and iterate our aggregation procedure until the selected context has not changed, or the maximum number of iterations have been reached.

On Egoschema, all context aggregation methods, including the single-step variant, improve over baseline inference. Egoschema videos are short (180 frames), and effectively fit into the model's context window. The fact that all approaches improve over the baseline emphasise the utility of curating the model's input context to remove distractors.

On LVBench, where videos are far longer (average of 68 minutes), our proposed Dynamic-Segment TCoT (with $l = 12$) shines. This approach is able to effectively consider the whole video whilst adhering to its 32K context limit per VLM call. Single-step TCoT does not improve over the baseline here; we posit this is due to processing only a sparse subset (120 of average of 4080) of the total frames, which is too few to identify the relevant frames. Hierarchical TCoT mitigates this problem by adopting a coarse-to-fine strategy, but it underperforms our partitioned aggregation as it may miss short events which are not present in the initial, coarse sampling of the video.

We used $s = 64$ frames, and $l = 12$ segments for our experiment, ablating this choice in App. A.

**Computation vs accuracy analysis**    We analyse the trade-off between computational cost and accuracy of TCoT in Fig. 4 on LVBench. We consider two alternatives: First, we perform standard inference with Gemini using a larger context limit. And second, as an alternate inference-time scaling strategy, we use "self-consistency" [59], where we sample multiple predictions from the VLM and take the majority vote as the final answer. These multiple results are obtained by randomly sampling 120 frames from the input, and also by increasing the sampling temperature to 0.7 [59] to obtain diverse output. We use the total number of tokens processed by the VLM to measure computation, as it is directly proportional to the monetary cost of using the model via an API. Moreover, metrics such as GFLOPs and inference-time are not available when calling the model through an API.

Figure 4 shows that as we increase the number of segments, $l$, and therefore the total number of tokens / frames processed, the performance of TCoT increases smoothly, whereas standard baseline inference saturates at around 1000 frames (264K tokens). When processing a total of 2700 frames (700K tokens), we are able to achieve an improvement of 2.8 points (61.7 vs 58.9) at the same cost.

As our approach curates the relevant context from the input video, and LVBench videos are very long with an average of 4080 frames, TCoT is not as effective when the total number of frames considered is low. This is because there are not sufficient input frames to select the most relevant context from. Therefore, we observe the benefits of our approach over the long-context baseline after processing a total of 512 frames (132K tokens).

Self-consistency chain-of-thought prompting [59], achieves minimal improvements over just a single inference call, highlighting that inference strategies developed for language are not directly applicable to video, which requires specialised approaches as ours.

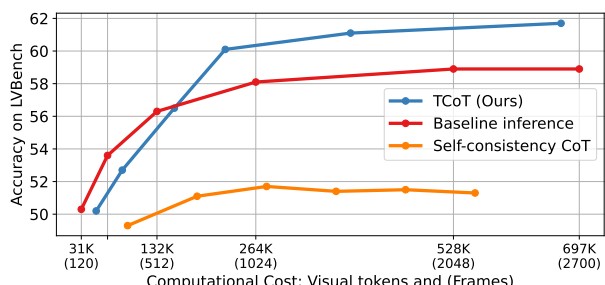

Figure 4: **Accuracy vs computation trade-off.** We compare Temporal Chain of Thought to two alternatives: baseline inference and self-consistency CoT [59]. We use the total number of visual tokens and frames (in parentheses) processed to measure computation, and vary $l$ from 2 to 32 to do so for TCoT. Our approach improves consistently whilst baseline inference saturates in the presence of distractors from more frames. Self-consistency CoT is ineffective, underlining the need for inference-time scaling approaches tailored to video.

Note that as we use $s = 64$ frames, TCoT does not exceed 32K input context tokens, regardless of $l$ and the total number of tokens / frames processed. Baseline inference, in contrast, is restricted by the context window supported by the model.

**Alternate context aggregation approaches.** Table 2 compares the following methods for choosing a fixed number of frames (120 to fit a 32K context limit) from a video:

- *Uniform sampling*: We uniformly select 120 frames.
- *Feature similarity*: We select frames based on their similarity to the embedded question using $k$-nearest neighbour search, as explored by [5, 16, 62]. We embed either the captions generated by our VLM ("question → captions") or the frames directly ("question → frames") using the strong dual-encoder SigLIP model [71]. For "question → captions", we prompt our VLM to generate concise captions to ensure they fit within the 64-token context limit of the SigLIP text encoder [71].
- *VLM-Based*: We use a VLM for selecting relevant context, either based on captions generated from the frames (as done by [21]), or by directly feeding the frames without any intermediate captions. We use the same partitioned segments (Sec. 3.2) in all cases.

We observe that all context aggregation approaches outperform naive uniform sampling on both datasets, highlighting the importance of curating the input to a VLM. Second, note that methods selecting frames directly with a VLM (rows 4–6) perform better than those relying on feature similarity (rows 2–3). Intuitively, this is because instruction-tuned VLMs can adapt easily to a new task in a zero-shot manner, and also dynamically vary the number of selected frames depending on the question type as shown in Fig. 6. Third, when selecting with a VLM, directly selecting from frames

Table 2: **Alternate context aggregation strategies**. We compare different methods for selecting 120 input frames. We use the same VLM, with 32K-context for the final answer.

| | Selection strategy | Egoschema | LVBench |
|---|---|---|---|
| 1 | Uniform sampling | 72.6 | 50.3 |
| | Feature similarity | | |
| 2 | Question → captions | 73.8 | 52.1 |
| 3 | Question → frames | 73.4 | 54.4 |
| | VLM-Based | | |
| 4 | Select from "concise" captions | 74.0 | 58.3 |
| 5 | Select from "long" captions | 72.8 | 60.4 |
| 6 | Select directly from frames (Ours) | **75.2** | **61.7** |
| 7 | Oracle with annotated time references | – | 67.4 |

as in our method (row 6) performs better than using intermediate frame captions which lose information. Moreover, captioning-based approaches are sensitive to the prompt used for captioning: "concise" and "long" captions perform differently for Egoschema and LVBench (rows 4 and 5). Our captioning prompts are detailed in App. D. Finally, we observe that when we use the oracle time-reference frames which are human-annotated for LVBench [58], our performance increases by 5.7 points. The largest headroom, however, is in improving the answerer model (ie VLM) which is not in the scope of this work.

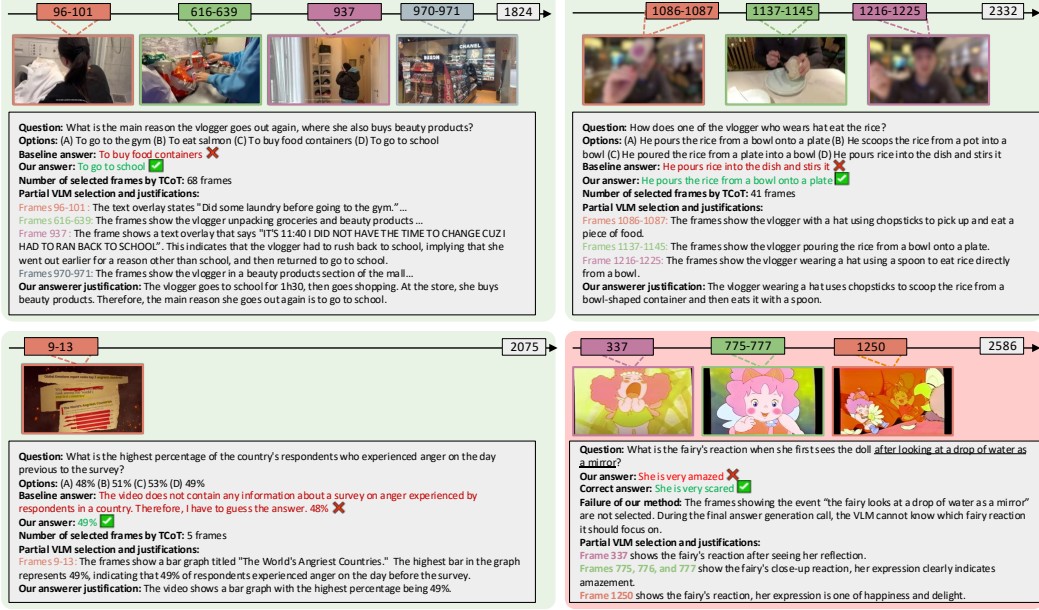

Figure 5: **Qualitative examples on LVBench.** Note how our model focuses on different parts of the video to make its prediction. For clarity, we sample frames from the segments selected by TCoT. In the failure case, although TCoT finds various frames showing the fairy's reactions, none of them include the drop of water, meaning that the answerer cannot make the correct prediction.

**Dynamic Context Aggregation** Figure 6 analyses the percentage of selected frames across different question types on LVBench. Observe how the proportion of selected frames adapts dynamically based on the question type. For example, temporal grounding questions focus on specific moments in time, and accordingly less than 10% of the frames are chosen. Summarisation leads to the highest proportion of selected frames, which intuitively agrees with the need

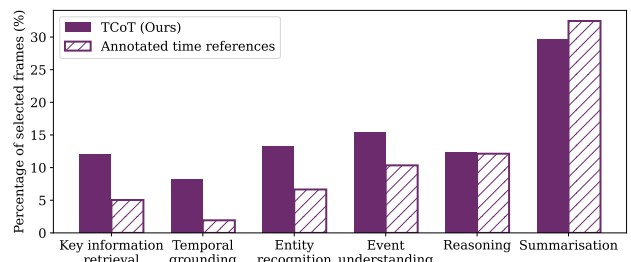

Figure 6: **Percentage of selected frames by question type.** Observe how the proportion of selected frames dynamically adjusts to the question type, aligning well with the human-annotated time-reference frames on LVBench [58].

for a broad understanding of the video. Note that our selected proportions are also in agreement with the proportion of human-annotated "time-reference" frames [58]. Appendix A shows that our selected frames show distinct behaviours across datasets, confirming that our approach is indeed adaptive. On Egoschema, we select a larger proportion of frames, which agrees with the fact that it was annotated by [41] to require looking at at least 56% of the video. Finally, App. A also analyses performance by question type, showing that TCoT shows the largest improvement over the baseline for "temporal grounding" and "key information retrieval" which requires precise localisation of relevant context.

**Qualitative examples** Figure 5 and App. C present both success and failure cases of our approach.

### 4.3 State-of-the-Art Comparison

Finally, we compare to the state-of-the-art on four video QA datasets in Tab. 3, focusing on long-video datasets with prior works leveraging VLMs.

**LVBench** We substantially outperform prior works on this recent, challenging dataset (Tab. 3) with an average length of 68 minutes. Note that our Gemini 1.5 Flash baseline that we used in our ablations (Sec. 4.2) was already state-of-the-art, and we achieved substantial improvements: Namely, we improve by 11.4 points with the same 32K context limit, and by 2.8 points when processing the same number of total tokens (700K) with our TCoT method.

To show the generalisability of TCoT to other VLMs, we also present results using Qwen 2.5 VL 7B [6] and GPT-4o-mini [1]. We run baseline inference for both models with the maximum number

Table 3: State-of-the-art comparison. We report our own Gemini 1.5 Flash baseline for Egoschema and LVBench as it outperforms [55, 58]. For LVBench and OpenEQA, we report the tokens used in a single context-window, and the total number of tokens processed. OpenEQA uses the "LLM-as-judge" protocol of using GPT-4 to evaluate the answer. †: Our reproduction.

**LVBench**

| Model | Context | Total | Accuracy |
|---|---|---|---|
| VideoAgent [16]† | – | – | 37.6 |
| InternVL-2.5-78B [12] | – | – | 43.6 |
| Qwen-2.5-VL-7B [6] | 128K | 128K | 46.1 |
| GPT-4o-mini [1] | 22K | 22K | 48.0 |
| Gemini 1.5 Flash [55] | 32K | 32K | 50.3 |
| Gemini 1.5 Flash [55] | 700K | 700K | 58.9 |
| TCoT (Qwen-2.5-VL-7B) | 128K | 320K | 49.1 |
| TCoT (GPT-4o-mini) | 22K | 86K | 53.5 |
| TCoT (Gemini 1.5 Flash) | 32K | 672K | **61.7** |

**Egoschema and Next-QA**

| Method | LLM / VLM | Egoschema Subset | Egoschema Full set | NeXT-QA Accuracy |
|---|---|---|---|---|
| LangRepo [28] | Mixtral | 66.2 | 41.2 | 60.9 |
| MoreVQA [42] | PaLM-2 | – | 51.7 | 69.2 |
| Video Agent [61] | GPT-4 | 60.2 | 54.1 | 71.3 |
| Video Agent [61]† | Gemini 1.5 Flash | 65.6 | – | – |
| LLoVi [72] | GPT-4 | 61.2 | 52.2 | 73.8 |
| GPT-4V [3, 7] | GPT-4V | 63.5 | 55.6 | – |
| LVNet [45] | GPT-4o | 68.2 | 61.1 | 72.9 |
| MotionEpic [17] | Vicuna | – | – | 76.0 |
| VideoTree [62] | GPT-4 | 66.2 | 61.1 | 75.6 |
| BOLT [38] | LLaVA-One | 60.6 | 64.0 | 79.5 |
| LongVU [49] | Qwen2-7B | – | 67.6 | – |
| Gemini Flash [55] | Gemini 1.5 Flash | 72.9 | 67.8 | 80.0 |
| TCoT (ours) | Gemini 1.5 Flash | **75.2** | **69.1** | **81.0** |

**Open EQA**

| Model | Context | Total | LLM Match |
|---|---|---|---|
| Claude 3 [4, 40] | 6K | 6K | 36.3 |
| GPT-4V [3, 40] | 4.2K | 4.2K | 55.3 |
| Gemini 1.5 Flash | 77.4K | 77.4K | 68.0 |
| TCoT (Gemini 1.5 Flash) | 32K | 76.4K | **69.2** |

of frames / tokens that it supports to obtain the strongest possible baseline. Namely, this is 1024 frames / 128K for Qwen 2.5 and 250 frames / 22K for GPT-4o-mini (the GPT API supports only 250 frames, even though it supports more text tokens in its context [1]). Our improvements with TCoT are consistent and considerable on these VLMs, especially since our iterative TCoT allows us to consider more frames than the model supports natively in its context window. In particular, our improvement over GPT-4o-mini is the largest, at 5.5 points, as it supports the smallest context window.

**Egoschema**    Table 3 shows that we outperform prior works on Egoschema. We compare to prior works that used either VLMs or LLMs (in conjunction with per-frame captioners) [45, 61, 62, 72]. We also re-implement VideoAgent [61] (details in App. D) using the same Gemini Flash VLM. Methods which initially compute captions on each frame are bounded by the quality of these captions, which often miss details relevant to the question. The Gemini 1.5 Flash baseline that we used in our ablations (Sec. 4.2) was already state-of-the-art, and we improve upon it further with TCoT.

**NExT-QA**    The videos in NExT-QA are the shortest from all of our evaluation datasets, and average only 39.5 seconds. Nevertheless, we use this dataset to compare to prior works on this dataset which have also leveraged VLMs or LLMs (operating on per-frame captions) in Tab. 3. There is less headroom for us to improve on this dataset, as the videos are short and the accuracy of our Gemini 1.5 Flash baseline is already high. Our results are still consistent with our other experiments and Egoschema. Namely, we outperform prior works and we reduce the relative error of our state-of-the-art Gemini 1.5 Flash baseline by 5.0%.

**OpenEQA**    OpenEQA presents a different domain, as the questions-answer pairs were labelled for mobile, embodied agents. Moreover, the answers are open-ended and an LLM (GPT-4 [2]), is used to evaluate answers in the authors' protocol [40]. Consistent with other datasets, we outperform prior works and our strong Gemini 1.5 Flash baseline using 300 frames (77.4K tokens).

## 5    Conclusion

We presented Temporal Chain of Thought, an inference strategy for long-video-question answering in VLMs, motivated by the fact that VLMs are affected by redundant information in their input context [23, 28, 37]. Our approach curates the input context of a VLM by decomposing a video QA task into first adaptively finding the relevant frames in the video. We demonstrated the efficacy of this approach by achieving state-of-the-art results on 4 different datasets. In particular, our approach with an input context of 32K tokens outperformed a 700K token model by 2.8 points on the challenging LVBench datasets where videos are 68 minutes long on average.

**Limitations**    Our approach, which we have shown to be effective across three different VLMs, nevertheless requires a model with good instruction-following capabilities in order to perform the selection function (Fig. 3) in a zero-shot manner. And whilst we have shown significant improvements from our inference strategy, future work could improve this further by training the model explicitly for this inference approach via reinforcement learning [20, 44].

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

## Broader Impact

Our work presents an inference-strategy to improve the performance of Vision Language Models (VLMs) on long videos (our improvements are the largest on hour-long videos). VLMs, and video question-answering, the domain of our work, is a generic technology with a wide range of potential applications. We are unaware of all potential applications, but we are cognizant that each use-case has its own societal impacts depending on the intentions and motivations of the individuals or organisations building and deploying the system. For example, long-video understanding can be used for productive purposes, such as a user asking a system detailed questions about long, instructional videos. However, it may also be used for applications such as surveillance systems.


Table 4: **Independent segment answer aggregation**: As an additional baseline to show that our method can reason across different segments, we partition videos into segments, answer questions independently within each window, and aggregate these answers to predict the final response. This baseline substantially underperforms our final method (Sec. 3.2) as it cannot reason across different segments.

| Model variant | LVBench |
|---|---|
| Individual segment answers | 51.0 |
| Individual segment answers with high confidence prompt | 55.6 |
| Ours | 61.7 |

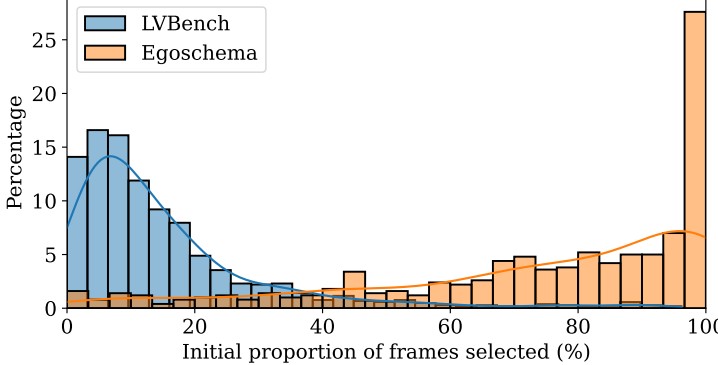

Figure 7: **Distribution of frames selected by our model**. Our model selects the relevant context in an adaptive manner, choosing a greater proportion of input frames for Egoschema (mean of 74%) than for LVBench (mean of 15%) The results on Egoschema correlate with its "temporal certificate" [41].

# Appendix

In this appendix, we include additional experimental results (App. A), detailed failure mode analysis (App. B), qualitative examples (App. C), as well as additional experimental details (App. D). References follow the numbering from the original paper.

# A Additional experimental results

**Independent segment answer aggregation**    To show that our approach is able to reason across different segments in the video, we perform the following experiment with an alternate baseline: We divide videos from LVBench of $N$ frames into $\lceil \frac{N}{s} \rceil$ segments of length $s$, and answer the question independently with a justification within each window. To compute the final answer, we pass each of the per-segment answers again to the VLM, and ask it predict the final answer. In Tab. 4, we observe that the performance is substantially lower on LVBench: 55.6% compared to our 61.7%. Intuitively, the final call to the VLM is often noisy because most segments are irrelevant. We found that the model should only propose an answer for a segment if it is highly confident ("high confidence prompt" in Tab. 4); otherwise, it attempts to answer for every segment, leading to contradictory and noisy information in the final VLM call. Additionally, this baseline underperforms compared to our method (Sec. 3.2) because many questions necessitate considering multiple segments. Therefore, our approach not only decomposes a video into smaller segments but also reasons across these segments, a capability lacking in simpler baselines.

**Distribution of selected frames.**    To further analyse the adaptivity of our method in selecting frames relevant to the question, Fig. 7 analyses the proportion of frames selected. Concretely, we plot the ratio of frames initially selected by TCoT (Eq. 5). We note that if the number of selected frames is larger than the context-limit of the model, $k$, we reduce it by uniformly sampling within the sorted list of frame indices (Sec. 3.2). Nevertheless, it is more informative to analyse this initial distribution, as it shows the frames which the model deems relevant.

Our selection function shows distinct behaviour across Egoschema and LVBench, confirming that it is indeed adaptive to the input questions. The proportion of selected frames is significantly higher on

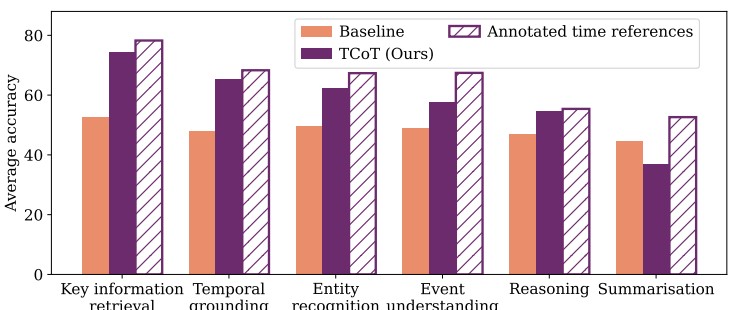

Figure 8: **Performance per question type on LVBench.** We compare baseline inference, our TCoT method, and using the oracle of the human-annotated time references to select relevant frames. We achieve significant improvements on most question types, and often near the accuracy of the oracle, particularly on "key information retrieval" and "temporal grounding" which requires precisely locating the relevant information in the long video.

Table 5: **Limits of Chain of Thought (CoT) methods in long video understanding**. Existing CoT techniques developed for language do not bring significant improvements on long video understanding. This calls for CoT techniques tailored to this task.

| Chain of Thought (CoT) technique | # VLM calls | LVBench |
|---|---|---|
| None | 1 | 49.5 |
| Zero-shot CoT | 1 | 50.3 |
| Zero-shot CoT with 2-stage prompting [29] | 2 | 49.4 |
| Zero-shot CoT with self-consistency [59] | 9 | 51.7 |

Egoschema, as its questions were designed by [41] to require a more holistic understanding of the video. Indeed the "temporal certificate" [41] (the duration of the video that an annotator must look at to answer the question) is estimated to be at least 100 seconds, or 56% of the video, which correlates with Fig. 7. Although LVBench videos are longer, their questions require localising specific moments in the video, as also shown in Fig. 7.

**Performance per question type on LVBench.** Figure 8 compares the performance of our method to baseline inference, and the oracle of using the annotated time-reference frames to select relevant frames. We observe that TCoT consistently outperforms baseline inference, particularly for "key information retrieval" and "temporal grounding" which requires precise localisation of relevant context. Moreover, we approach the accuracy of the oracle on these categories too. However, we note that baseline inference performs better for summarisation questions, which require a holistic view of the video. This is also the only question type in Fig. 6 where our method on average selects fewer frames than the oracle, suggesting that we do not select enough relevant information in these cases.

**Limitations of pure text CoT on long video reasoning.** Chain of Thought (CoT) prompting techniques enhance the reasoning capabilities of LLMs across various tasks. Inspired by these successes, we explore whether typical CoT methods can also improve performance in long video understanding in Tab. 5. In particular, we investigate:

- Zero-Shot CoT [29]: We add a sentence encouraging step-by-step thinking, namely "Explain your reasoning," into the prompt before outputting the final answer (Fig. 14).

- Zero-Shot CoT with two-stage prompting [29]: we append the output of the initial Zero-shot CoT to the prompt and generate a new prediction.

- Zero-Shot CoT with self-consistency [59]: we set the VLM temperature to 0.7 and sample a different set of frames at each run to encourage diverse reasoning across runs. The final prediction is determined by majority voting from nine runs, each employing Zero-Shot CoT. We experimented with more runs but observed no performance improvement (shown in Fig. 4).

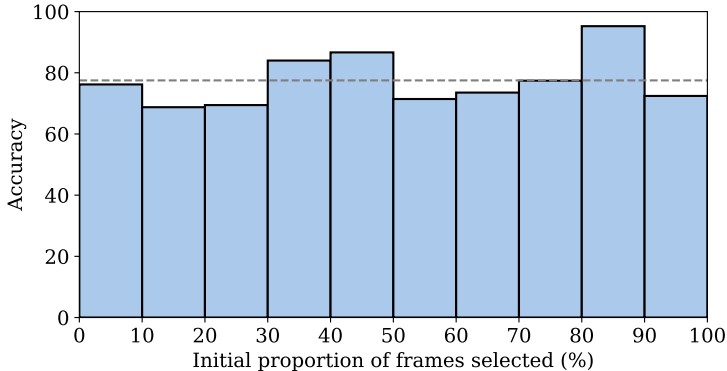

Figure 9: **Accuracy according to the proportion of frames selected on Egoschema**. Observe how our model's accuracy remains consistent, regardless of the number of frames selected, suggesting that our TCoT method can effectively and adaptively select the relevant frames to answer the question. The dashed line shows the overall average accuracy on the dataset.

Table 6: **Effect of hyperparameters**. We analyse the effect of the segment size, $s$ (a), and the number of uniform context frames, $u$ (b). The context-limit is $k = 120$, meaning that the remaining $m = k - u$ frames are selected by the model.

<table>
<tr><td colspan="3">(a) Effect of segment size, $s$</td><td colspan="4">(b) Effect of uniform context, $u$.</td></tr>
<tr><td>$s$</td><td>Egoschema</td><td>LVBench</td><td>$m$</td><td>$u$</td><td>Egoschema</td><td>LVBench</td></tr>
<tr><td>4</td><td>73.0</td><td>56.8</td><td>120</td><td>0</td><td>**75.2**</td><td>57.8</td></tr>
<tr><td>16</td><td>73.6</td><td>57.0</td><td>88</td><td>32</td><td>74.6</td><td>58.5</td></tr>
<tr><td>32</td><td>73.8</td><td>**58.1**</td><td>64</td><td>56</td><td>73.2</td><td>**59.3**</td></tr>
<tr><td>64</td><td>**75.2**</td><td>57.8</td><td>32</td><td>88</td><td>74.0</td><td>56.2</td></tr>
<tr><td>120</td><td>73.8</td><td>57.8</td><td>0</td><td>120</td><td>72.6</td><td>50.3</td></tr>
</table>

In Tab. 5, we observe that all the pure linguistic CoT techniques result in only marginal performance improvement on LVBench. This result calls for VLM inference strategies specifically tailored to video understanding tasks such as ours. Finally, note that we adopt "Zero-Shot CoT" as our default prompting technique for the VLM call generating the final answer (Fig. 14).

**Accuracy is consistent across selected frames**    To further analyse the quality of our frame selection, Fig. 9 compares the accuracy of our TCoT model as a function of the proportion of frames initially selected.

If a model is effective in adaptively selecting only the frames that it needs to answer the question, then its accuracy should remain constant regardless of the number of frames selected. Figure 9 shows that this is largely the case for our TCoT model, highlighting that our approach effectively selects the frames it requires for the task.

**Effect of hyperparameters, $s$ and $u$**    Table 6 shows the effect of TCoT hyperparameters (Sec. 3.2), $s$ (segment size) and $u$ (uniform context).

Table 6a shows that the performance of our model is relatively consistent across different segment sizes, $s$, on both Egoschema and LVBench datasets. A smaller segment size means that we require a smaller context window during context aggregation, whilst also requiring more VLM calls. Note that the total context-limit to the answerer, remains constant at $k = 120$ here, enabling accurate predictions even with a small window, $s$.

Table 6b varies the amount of uniform context, $u$, that we add. Once again, the total context-limit remains fixed at $k = 120$, meaning that the remaining $m = k - u$ frames are selected by the model. Moderate amounts of uniform context help on LVBench, as they enable the model to obtain a broader awareness of the video. Egoschema, in contrast, does not benefit from uniform context, and this may be explained by Fig. 7 which shows that the model is already selecting a larger number of frames on this dataset, which has a "temporal certificate" [41] which covers the majority of the video. Finally, note that when $m = 0$, we are effectively performing the standard inference baseline as no frames are

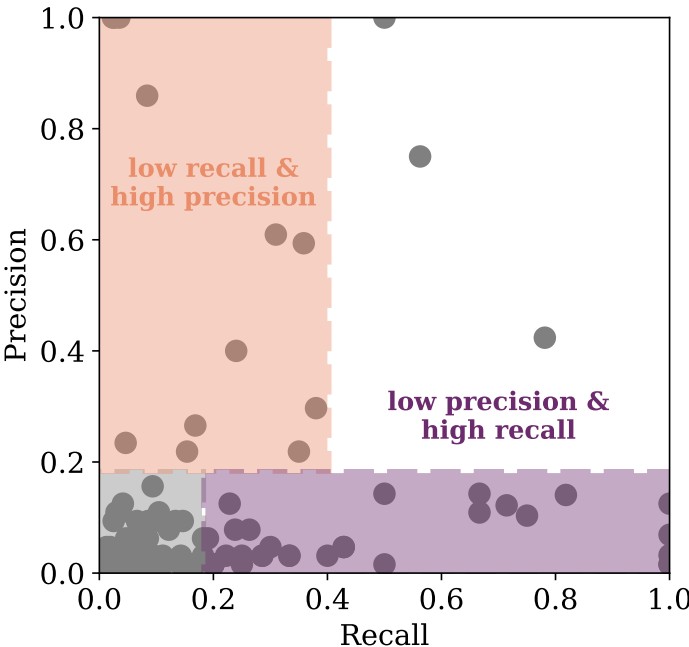

Figure 10: **Failure modes of TCoT** We show the precision and recall of frames that have been aggregated by our method, compared to the human-annotated time-reference frames of LVBench. This is performed for the instances in the dataset where TCoT fails, while the oracle which uses only the time-reference frames for answering succeeds.

being selected by the model. Overall accuracy degrades, particularly on LVBench, as discussed in Tab. 1.

## B    Failure mode analysis

We examine the failure modes of our Temporal Chain of Thought in detail in Fig. 10. To do so, we use the annotated "time reference" segments from LVBench [58], which are segments of video that contain the necessary information to answer the question accurately. Ideally, assuming that the time-reference annotations are completely correct, we should only select the time-reference frames and no other ones.

In Fig. 10 we evaluate the precision and recall of our selected frames, where a Frame ID is denoted as a "true positive" if it falls within the annotated time reference segment and "false positive" if it does not. High precision means that few selected frames fall outside the time reference frames, while high recall indicates that most of the time reference frames are included in the selection. In Fig. 10 we plot only the 168 instances where TCoT provides *incorrect* answers to the question, while the oracle answers correctly.

We identify prevalent failure modes in Fig. 10 and show qualitative examples for these modes in Fig. 11:

- **Low precision and high recall** (purple area in Fig. 10 and qualitative examples in Fig. 11a). The model selects too many frames, selecting frames that are only remotely related to the question. This over-eagerness leads to excessive and irrelevant frame selection. For example, we see in Fig. 11a (left) that if the question asks what the dragon is dropping while in the sky, the method selects frames of the dragon even when it is clearly not in the sky. Our model sometimes struggles with questions that require understanding connections between segments, such as determining the second-to-last event in a video as shown in Fig. 11a (right).

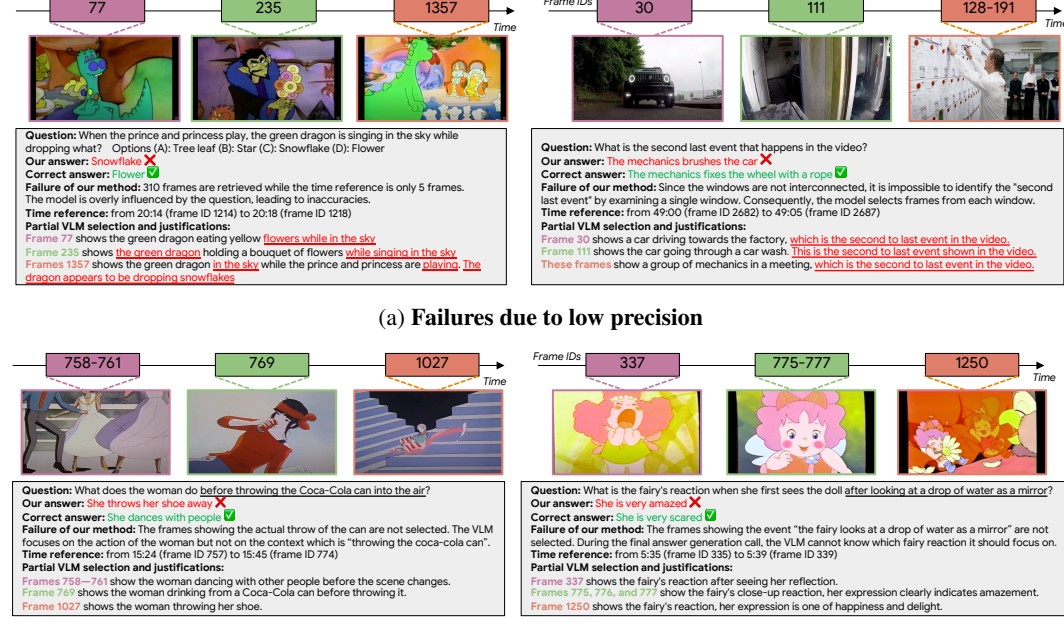

(a) **Failures due to low precision**

(b) **Failures due to low recall**

Figure 11: **Qualitative analysis of the failure modes of Temporal Chain of Thought. Failures due to low precision:** Sometimes, our method selects too many frames. In the top left example, it is overly influenced by the question and selects frames at each window, even when they are not highly relevant. For instance, if the question asks what the dragon is dropping while in the sky, the method retrieves frames of the dragon even when it is clearly not in the sky, incorrectly assuming it is. In the top right example, it is impossible to determine the "second last event" of the video by looking at a single window. As a result, the model incorrectly tries to answer the question within each single window. **Failures due to low recall:** Sometimes, our method misses important frames, such as those showing "throwing the Coca-Cola can" (left) or "looking at a drop of water as a mirror" (right). These missing frames are essential for accurate final inference.

- **Low recall and high precision** (orange area in Fig. 10 and qualitative examples in Fig. 11b): The selected frames are relevant, but the model misses crucial parts of the question. For example, in Fig. 11b (right) in response to the question, "What is the fairy's reaction when she first sees the doll after looking at a drop of water as a mirror?", TCoT selects frames of the fairy's reaction but omits frames showing the fairy looking at the water drop. Consequently, the final answer is inaccurate because the final VLM call cannot determine which of the fairy reactions it sees occurs after looking at a drop of water.

## C  Qualitative results

Figure 12 visualises additional examples of our TCoT method on LVBench [58], following the same format as Fig. 5 in the main paper.

Figure 13 shows an example of why adding uniform context (described in Sec. 3.2 of the main paper) is beneficial.

## D  Additional experimental details

**Complete prompts**  Figure 3 in the main paper showed our VLM selection call, $S$ (Eq. 4). For completeness, we also include the prompt for answering, $H$, (Eq. 3 of the main paper) too. Specifically, Fig. 14 and 15 shows the prompt for multiple-choice questions (Egoschema [41], LVBench [58] and NeXT-QA [66] datasets), and Fig. 16 for open-ended questions (for the OpenEQA [40] dataset).

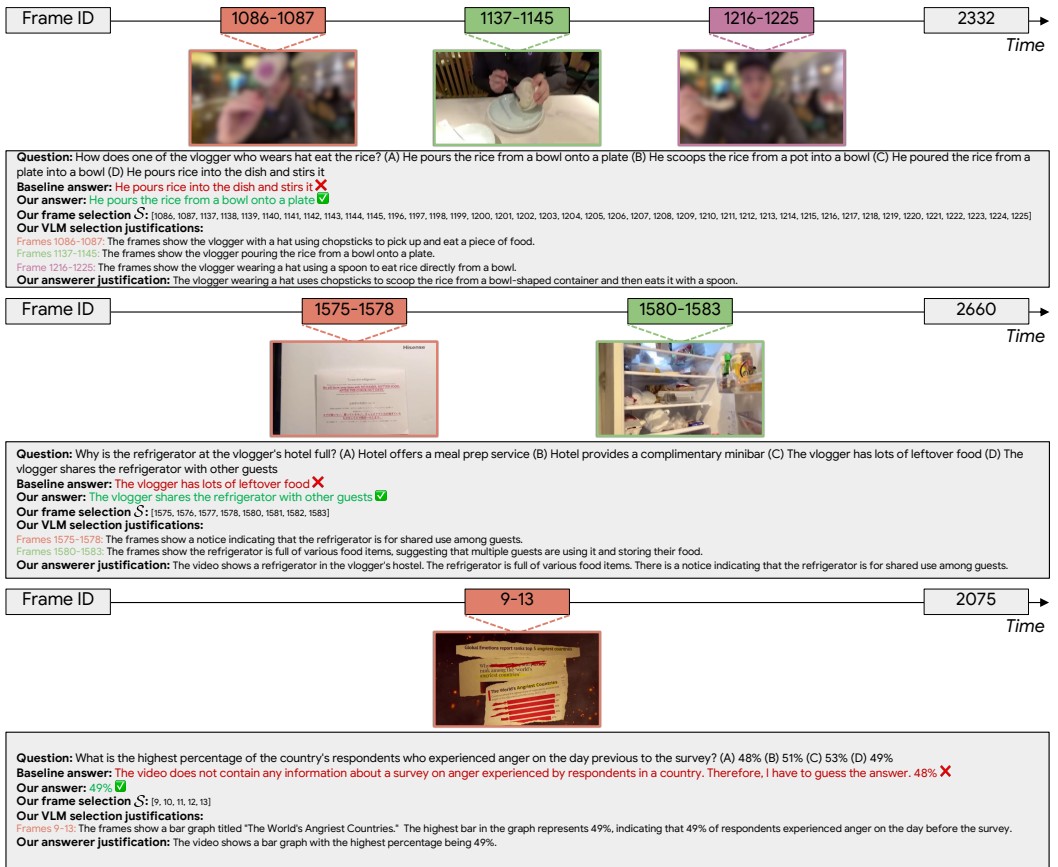

Figure 12: **Additional qualitative examples on LVBench.** Note how our model focusses on different parts of the video to predict the correct answer. The top row shows an example of the model focussing on 3 diverse segments of the same video. The second row includes two such segments. The third row shows an example of how our TCoT approach is able to find the 4 frames in the entire video of 2075 frames which contain the correct answer. In all cases, for clarity, we show a single frame from each selected segment of frames. Faces have been blurred in the first row.

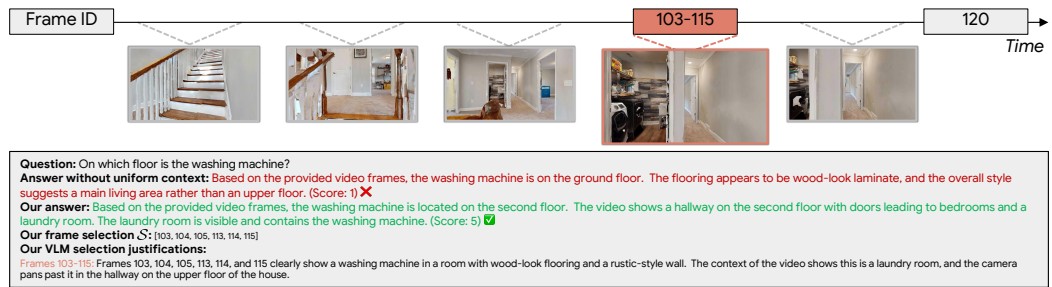

Figure 13: **Importance of adding uniform context** Our TCoT approach is able to select the relevant frames (103 to 115) for the question, by focussing on the washing machine (and provides the correct justification for doing so too). However, if we pass only these selected frames to the answerer, the result is incorrect as relevant information about which floor the machine is on is lost. To alleviate this issue, we therefore include uniformly sampled context as well, as visualised in this figure, and described in Sec. 3.2 of the paper. Example from OpenEQA [40].

```
You will be given a question about a video and {number of choices} possible answer
options. You are provided frames from the video, retrieved by an intelligent agent.

Frames: {frame1}, ..., {frame N}
Question: {question}
Possible answer choices: {answer choices}

After explaining your reasoning, output the final answer in the format.  "Final
Answer: (X)" where X is the correct digit choice. Never say "unknown" or "unsure", or
"None", instead provide your most likely guess.
```

Figure 14: **Multiple choice question prompt for the Gemini and GPT-4o-mini answering call**, $H$ (Eq. 3).

```
Frames: {frame1}, ..., {frame N}

Carefully watch the video and pay attention to the cause and sequence of events,
the detail and movement of objects and the action and pose of persons. Based on your
observations, select the best option that accurately addresses the question.
Question: {question}
Options: {answer choices}
Answer with the option's letter from the given choices directly and only give the best
option.
```

Figure 15: **Multiple choice question prompt for the Qwen 2.5-VL answering call**, $H$ (Eq. 3).

**Hierarchical TCoT**    We include further details of hierarchical aggregation, which was introduced in Sec. 4.2 of the main paper.

We hierarchically extend Single-Step TCoT (Sec. 3.2) by iteratively sampling around the previously identified frames of interest. Intuitively, in a long video where our model's context limit is far smaller than the number of input frames, if we first coarsely sample frames, we may miss many necessary frames. Therefore, once we have identified frames relevant to the question, we select nearby ones that were not considered initially and iterate.

Concretely, the first iteration follows "Single-Step TCoT" (Sec. 3.2): Given a set of frames, $\mathbf{x}_0$ sampled uniformly from the video, we identify relevant frames, $\hat{\mathbf{x}}_0, \mathcal{S}_0 = S(\mathbf{x}_0, \mathbf{q})$, where $\hat{\mathbf{x}}_0 = \{x_i, \ldots, x_j\}$ for $i, j \in \mathcal{S}_0$. Subsequently, we "zoom in" on these relevant regions by sampling additional frames around $\hat{\mathbf{x}}_0$ (Fig. 2b). Specifically, for each index in $\mathcal{S}_0$, we construct a neighbourhood of frames, $\mathbf{x}_1 = \{x_{i-v}, \ldots, x_i, \ldots, x_{i+v}, \ldots, x_{j-v}, \ldots, x_j, \ldots, x_{j+v}\}$, where $v$ denotes the neighbourhood size, and duplicate frames are removed.

The new sequence of frames, $\mathbf{x}_1$, serves as the input for the next iteration. We repeat this algorithm until convergence, namely until $t$ iterations have been completed or earlier if the relevant set of frames has not changed, or in other words, $\hat{\mathbf{x}}_{i+1} = \hat{\mathbf{x}}_i$.

**Video Agent reimplementation**    We reimplemented the Video Agent framework [61] with several modifications to make it directly comparable to our approach. In particular, Video Agent used LaVila [73] as its captioner, whilst we use Gemini 1.5 Flash for this purpose for fair comparison. Similarly, instead of using EVA-CLIP-8b-plus [53], we use SigLIP-So400m/14 [71] as the other baselines in Tab. 2. Finally, we used Gemini 1.5 Flash as the answerer to be consistent with the rest of our work, instead of the original GPT-4. Video Agent starts off with an initial number of uniformly selected frames. We ablated this hyperparameter and achieved the best performance with 50 initial frames on Egoschema, and 64 initial frames on LVBench.

```
Frames: {frame1}, ..., {frame N}

You will be given a question about a video.You will be provided frames from the
video, retrieved by an intelligent agent.  It is crucial that you imagine the visual
scene as vividly as possible to enhance the accuracy of your response.
Question: {question}
```

Figure 16: **Open-ended question prompt for our Gemini answering call**, $H$ (Eq. 3).

**Details of captioning for feature-similarity based aggregation** An additional baseline that we used in the paper was retrieve frames based on the feature-similarity to their captions (Tab. 2). Here, we include additional details of it.

We generate two types of captions for each frame of the video frame using Gemini 1.5 Flash. For *concise* captions, we use the prompt *Write a concise description of the image in a sentence*. For *long* captions, we use the prompt *Write a paragraph describing the image in detail*. These captions were then embedded using the most performant version of SigLIP [71], SigLIP-So400m/14. We then used these embeddings to perform nearest-neighbour search, selecting the most similar captions to the input questions. The frames corresponding to these captions were then used to aggregate the context, **c**, that was then passed to the answerer VLM.

**Computational Resources** We call the Gemini 1.5 Flash [55] and GPT-4o-mini [1] through their public APIs. Qwen-2.5-VL [6] has publicly-available weights, and we run the HuggingFace implementation using a server with 8x NVIDIA A100 GPUs.

**Video IDs from LVBench** The videos in LVBench [58] are originally from YouTube. Since some of these videos are no longer available, we include the full list of Video IDs that we were able to obtain below.

Each video has on average 15.1 questions associated with it, for a total of 1043 questions.

The full list of all Video IDs that we were able to obtain are:

- -WnyRMZqV1U
- 16Z-XQh9jhk
- 2LH3JCGkEBU
- 2sriHX3PbXw
- 2zkJFv-ro4A
- 3_upA09AntU
- 4LA_tH-VSnQ
- 81SbCR6p3Z0
- 9-gOCOu_KGU
- 9tBsMSDoDqk
- AeEYQ62t8hA
- CgvJqGxzRfE
- Cm73ma6Ibcs
- EwskdNETNx8
- FaV0tIaWWEg
- HfEVEGf1A8Q
- JPPMz8fEml0
- JTa_Ue2MSwc
- JlrzSvCsIjE
- KbahC-QCKU8
- Mcggugol2ts
- NzCO0G8AGLU
- O14bbpvy2x0
- QB7FoIpx8os
- QgWRyDV9Ozs
- RCAqKnvu_P0

- RjDrZkBwzho
- S8vPx-u9p_A
- SRq0weUKskM
- TJR1oYDDTwg
- TZ0j6kr4ZJ0
- VTCDQYYKA9o
- Va_9Q6ekm60
- Vk_Af0htZGU
- W-BnDvXXfOs
- XNtNNplAwiI
- Xjf5N9S3jAA
- YlQugR7KSKg
- Z4HGQL_McDQ
- Z86xysw5Ncc
- Za2Z_JRxCuk
- _T2Avd3tFHc
- aJI8XTa_DII
- cWEnogdsW78
- cXDT44zT8JY
- gbDR39yIs3Y
- hROKtPqktO8
- hjoDzK0siaM
- iA_69g87Ilw
- ihfjEFGdZdc
- jp2M1hIEtsk
- k2FIFQIYBvA
- lDlA7cfNk8A
- o-gLbgpzCc8
- pXD3txG2bVQ
- q01CUy_gwdU
- qAIRFyR6NyQ
- qYMnM5blZIE
- rk24OUu_kJQ
- rp4NKWb7dXk
- sk00epALZps
- t-RtDI2RWQs
- tH_5YbklevQ
- tKIFQI9cH2c
- uW9mcG0rdLY
- vHlSoxg8WHo
- vaL_vSdZKZo
- wgBlACG927Y
- xi6r3hZe5Tg

