# OpenReview forum: "Temporal Chain of Thought: Long-Video Understanding by Thinking in Frames"
_NeurIPS.cc/2025/Conference — NeurIPS 2025 poster_

### Official Review · Reviewer_HQTU · 2025-06-15

**Clarity:** 3
**Significance:** 2
**Originality:** 2
**Rating:** 3
**Confidence:** 4

**Summary:**

This paper presents "Dynamic Context Aggregation" an inference strategy for long video question-answering (QA). The proposed method tackle a problem of current VLMs, that despite the current ability of these models to process long contexts as input, they still struggle with distractors in the input sequence, affecting the performance in some tasks. The proposed approach tackle long VQA by first selecting relevant frames from the video (frame id + text justification) and then using this information to answer a given question. The method achieves state-of-the art results on 4 video question-answering datasets.

**Questions:**

Please refer to weaknesses for my questions and doubts. I'm currently giving a rejection to this paper. Now I do not see a strong enough contribution or really new insights. After the rebuttal and the response from the authors I will revise my decision.

**Ethical Concerns:**

["NO or VERY MINOR ethics concerns only"]

**Final Justification:**

As I expressed in my response to the authors and in the "Questions" section of my original review, my main point about the paper is related to novelty, I don't see strong enough contributions from the paper to be accepted at NeurIPS. Therefore, I'm maintaining my original score of borderline reject.

**Limitations:**

yes

**Quality:**

2

**Strengths And Weaknesses:**

Paper Strengths:
1. The paper proposes Dynamic Context Aggregation, a combination of existing techniques that shows to be effective for selecting relevant information from videos, which shows to be useful for long video QA.
3. Experiments show that Dynamic Context Aggregation outperforms other methods in 4 different datasets, achieving SOTA results.
4. Most sections are well written and easy to follow

Paper Weaknesses:
1. The paper states that sometimes relevant frames are too succinct, which makes it for difficult for the second stage of the method to give a correct answer. Thus, it was found that is beneficial to include coarse uniform context (uniformly sampled frames) from the original input in addition to the selected frames. However, I’ve not been able to find this experiment to see the performance gain that is coming from adding this extra context and see if the effectiveness is coming from the good frame selection of the proposed method or from this additional context.
2. I think that the paper has a bit of contradictory ideas, first the paper claims the well know fact that too much information affect VLMs, therefore a method to select the most important information is proposed, however the proposed method select important frames and then again add more frames from the entire video to give more context to the VLM. Maybe a better description can help clarifying this and avoid this appearance of contradictory ideas.
3. The comparison with related works states that other works are not conditioned on the input question, and that the proposed method is simpler and elegant due to the use of a single VLM and do not rely on any external models or tools, however this approach is not entirely new, personally I’ve seen this idea of using a single VLM to get useful information by conditioning the model on the input question and provide an answer for Video QA, for example:
[1] https://aclanthology.org/2024.findings-acl.555.pdf

---

> ### Author Rebuttal · Authors · 2025-07-30
>
> We thank the reviewer for their detailed comments and address each concern below:
>
> ---
> ### **Ablation of uniform context**
>
> Thank you for this suggestion. We already included an ablation of uniform context in Table 6b of Appendix A. In the revision, we will refer more clearly to this experiment.
>
> For convenience, we have copied Table 6b below. We ablated the effect of the number of uniform context frames, $u$, on both Egoschema and LVBench. In both cases, the context-limit is $k = 120$ frames, meaning that the remaining $m = k - u$ frames are selected by the model.
>
> | m | u | Egoschema | LVBench |
> |---|---|-----------|---------|
> | 120  | 0  |  **75.2**         |  57.8       |
> | 88  | 32  |  74.6         |  58.5       |
> | 64  | 56  | 73.2          | **59.3** |
> | 32  | 88  | 74.0          | 56.2 |
> | 0  | 120  | 72.6          | 50.3 |
>
> On LVBench, we achieve moderate improvements (+1.5 points) by introducing uniform context frames (first row vs third row). If we perform no frame selection, and only use uniformly sampled frames, the accuracy is substantially worse (decreases by 9 points) emphasising that good frame selection is key to good performance.
>
> Egoschema, in contrast, does not benefit from uniform context. This may be explained by Figure 7 of Appendix A, which shows that our approach is already selecting an average of 74% of the input frames in the dataset (and we therefore do not need any additional frames). The high proportion of selected frames is in agreement with the dataset’s “temporal certificate” – Egoschema was annotated [40] such that the annotator must look at, at least 100 seconds or 56% of the video to answer the question.
>
> Finally, note that the Accuracy vs Computation analysis in Figure 4 accounts for the uniform context in our method, and it clearly shows the benefit of our method over baseline inference (which is effectively only uniformly sampling frames from the input video).
>
> ---
> ### **Motivation for uniform context**
>
> Intuitively, uniform context is beneficial for long videos (such as the hour-long videos in LVBench), since during context aggregation, we extract relevant context from each segment of the video independently. Adding a few coarsely sampled, uniform context frames, provides a more holistic view which improves accuracy. This is not required for shorter videos, such as those in Egoschema, which may already fit within the context-limit of the model.
>
> We will clarify this in the revision.
>
> ---
> ###  **Other methods which use only a single VLM for context selection**
>
> Thank you. We will include a discussion of [A] in the revision, and also revise the Related Work section, specifically the “Long-context video understanding” paragraph in light of [A]. Although [A] uses a single VLM, they first compute captions as an intermediate representation of the video, and select frames based on these captions. We considered this design choice too, and ablated this in Table 2, Rows 4 and 5. Our method, based on selecting relevant frames, achieves higher accuracy than selecting frames based on their captions. Moreover, it is not as sensitive to the captioning prompt that was used, as we observed that Egoschema and LVBench require different lengths of captions to achieve high results (detailed captioning prompts are in Appendix D, Lines 1093-1094). For convenience, we have reproduced the relevant parts of Table 2 below here.
>
> |                                    | Egoschema | LVBench |
> |------------------------------------|:---------:|:-------:|
> | Select from "concise" captions     |  74.0          |  58.3       |
> | Select from "long" captions        |    72.8      |   60.4      |
> | Select directly from frames (ours) |   **75.2**        |   **61.7**      |
>
>
> These results show that the general principle of curating the input to the VLM is beneficial, and that our instantiation of Partitioned Context Aggregation is the most effective.
>
> [A] D Romero et al. Question-Instructed Visual Descriptions for Zero-Shot Video Question Answering. ACL 2024

---

> ### Comment · Reviewer_HQTU · 2025-08-05
>
> Thanks to the authors for answering my questions. I have read the rebuttal and also the responses from the other reviewers , regarding the "Ablation of uniform context" I believe that a demonstration in two datasets where it improves in one of them and not in the other is not really convincing, and the paper would benefit with additional experiments to really show the readers the importance of this part (I understand that maybe the authors did not have enough time to perform extensive demonstrations for now). On the other hand, thanks for the responses regarding my other two points. But despite all of this, as I expressed in the "Questions" section of my original review, my main point about the paper is related to novelty (which has been also raised by buo9), key frame extraction has been very well established and the idea of selecting frames to improve performance has been around for a while, and despite achieving higher performances and some differences with other works, I don't see the additional points of the paper like the inclusion of additional context, as strong enough contributions that other people can benefit from to progress on this task.

---

> > ### Author Response · Authors · 2025-08-06
> >
> > Thank you for your constructive feedback.
> >
> > **Novelty** We emphasise that although frame extraction methods are well-established, we are the first method to our knowledge to do this directly with VLMs. This is an important distinction, because traditional methods cannot be applied to state-of-the-art VLMs (they are trained for each answerer model, and this would require backpropagating gradients through models like Gemini or GPT-4o which is infeasible) and are also designed for a fixed number of input frames, like 32 or 64. Therefore, they clearly do not scale to the 1000’s of video frames which we have demonstrated here. Further details are in our response to Reviewer buo9 and Lines 106-111.
> >
> > Our Partitioned Context Aggregation method (Sec 3.2) also allows us to extract relevant context from videos of any length, using a fixed context-window limit of the VLM. Exhaustive experiments (Tables 1 to 3, Fig 4, 6, Appendix A) also show that this method is more effective than other alternatives, and previous methods in the literature based on initial per-frame captions.
> >
> > Furthermore, although our method is simple, the insight behind it is not trivial, as evidenced by the fact that our inference strategy has not been proposed before in the literature. Moreover, our improvement of up to 11.4 points / 22.6 % relative on the most challenging, hour-long videos on LVBench is a significant result. Our improvements are consistent on 5 datasets and 3 state-of-the-art VLMs.
> >
> > Finally, since our method is simple and easy to understand and implement, it can readily be adopted by the community as mentioned by Reviewer eVAj.
> >
> > **Uniform Context** We will clarify further in the revision that the efficacy of uniform context is dataset-dependent. On long-video datasets such as LVBench, which have an average of 4080 frames at 1 fps, it is beneficial to add a few coarsely sampled, uniform context frames to provide a holistic view to improve accuracy. On shorter datasets, such as Egoschema, which contains exactly 180 frames at 1 fps and already fits within the context window of the model, it does not provide additional benefits.
> >
> > We also emphasise that the primary novelty of our approach is from using the VLM itself to perform frame selection, and our method of Partitioned Context Aggregation which allows us to process videos of any length with a fixed context-window limit of the model, and results in substantial accuracy improvements.

---

### Official Review · Reviewer_buo9 · 2025-06-25

**Clarity:** 1
**Significance:** 3
**Originality:** 3
**Rating:** 5
**Confidence:** 4

**Summary:**

The paper proposes a "Chain of Thought" like approach for sampling long for video for video understanding.
The authors propose to use VLM for context aware frame selection as opposed to uniform sampling in long for videos.
The paper shows strong reasons fordoing so in terms of performance results and multiple ablation studies and analysis of them.
While particular, The paper introduces a segmented approach to video understanding where the frames are first divided into non-overlapping segments and frames are chosen through a VLM. These selected frames are then used to answer the question.

While the work shows strong reasons to use the method when it comes to performance. Their analysis of performance vs computation falls short. While the authors do present some findings, it is difficult to understand as to how their method would affect efficiency in the first place as the number of frame processed by the LLM remains the same. Second, they seem to very the number of segments as a way to increase computation but it remains unclear as to how this would be the case.

**Questions:**

Please clarify/fix the weaknesses as mentioned above.

**Ethical Concerns:**

["NO or VERY MINOR ethics concerns only"]

**Final Justification:**

The low score was due to the concerns as portrayed on the review.

Given the rebuttal, I can conclude that I had misunderstood the sole parts of the working which have been clarified during the rebuttal.

With this, I believe the impact of the work greater than what was previously given.

**Limitations:**

yes

**Quality:**

3

**Strengths And Weaknesses:**

Strengths:
1. Paper shows a simple and elegant approach to a difficult problem.
2. Despite being simple, the paper is backed by strong improvements while being applicable in various areas.
3. Strong ablation study of their own methodology covering different types of segmentation.

Weaknesses:
1. While the performance is strong, the idea of key frame extraction has been very well established. The authors do no compare their work with widely used key frame extraction methodology and only focus on mainly LLM counterparts.
2. The efficiency argument seems unclear and even misguided as it is difficult to understand how the number of tokens being processed change at all as the authors process $m.l$ frames where $m=N/l$. Thus, I would like some clarification about the same.
3. The claim of constant time compute of videos regardless of the number of frames as mentions in the Discussion section in section 3 seems largely false due to the same reasons as mentioned in weakness 2.

---

> ### Author Rebuttal · Authors · 2025-07-30
>
> We thank the reviewer for their detailed comments and address each concern below:
>
> ---
> ### **Difficult to understand how the total number of tokens being processed change**
>
> We would like to clarify some confusion about the total number of tokens / frames that are processed.
>
> In Partitioned Context Aggregation (L174 to 198), the reviewer is correct that a video of $N$ frames is partitioned into $l$ segments, each of length $m = N / l$ frames.
>
> However, from each segment, $\\mathbf{x}\_{i}$, we uniformly sample $s$ frames from it (L182 to 185), denoted as $\\mathbf{x}\_{i [s]}$ in Equation 5 (*we believe this is the key detail that the reviewer missed*).
>
> Therefore, we process a total of $s \cdot l$ frames. Note that both $s$ and $l$ are independent of the number of the video length, $N$.
> This allows us to control the total amount of computation used to process a video in Figure 4, where we keep $s = 64$ frames constant, and vary $l$ from 2 to 42.
> Finally, note that for shorter videos where $s \cdot l > N$, or in other words, the video is too short to partition into $l$ segments, we set $l = \lfloor N / s \rfloor $ instead.
>
> We will revise the text to clarify this, and are happy to engage with the reviewer during the discussion period if this is not clear.
>
>
> ---
> ### **How is compute constant regardless of the number of frames?**
>
> As described above, we process a total of $s \cdot l$ frames, where both $s$ and $l$ are hyperparameters, and do not depend on the video length, $N$.
>
>
> ---
> ###  **Authors compare to LLM-based frame extraction methods, but not traditional ones.**
>
> Thank you for this suggestion. As we discussed in Lines 106-111, traditional frame selection models are trained to process only a fixed, small number of input frames (ie 32 for SeViLA [68] and ViLA [59], and 64 for [A]). These models are therefore not suitable for hour-long video datasets such as LVBench which contain an average of 4080 frames.
>
> Moreover, these frame-selection models are trained for specific downstream answerer models, and need to backpropagate gradients through this downstream answerer during training. It is therefore not feasible to train such approaches for large, state-of-the-art answerer models like Gemini 1.5, GPT-4o and Qwen-2.5 VL which we use in our work.
>
> We will clarify the limitations of traditional frame-selection techniques even further in the revision.
>
> [A] S Buch et al. Flexible Frame Selection for Efficient Video Reasoning. CVPR 2025.

---

> > ### Comment · Reviewer_buo9 · 2025-08-02
> >
> > Thank you for the detailed response clarifying my questions.
> > I believe I have a clear understanding now and see the impact of the work.

---

### Official Review · Reviewer_eVAj · 2025-06-28

**Clarity:** 3
**Significance:** 3
**Originality:** 2
**Rating:** 4
**Confidence:** 4

**Summary:**

LLMs (and consequently VLMs) have been shown to degrade for longer inputs in “needle-in-the-haystack” scenarios of question-answering, where only a small portion of the input is relevant while the rest is considered distractions. The paper suggests a simple method to curate long input videos for VLMs by prompting the VLM to select key frames for the given input question. The authors show that the curated input improves results on short inputs (which fit inside the context window), effectively removing distractions from the input. In addition, they show that feeding this input to a small context-window model performs better than a large context-window model when fed with the entire video.

**Questions:**

See the strengths and weaknesses section.

**Ethical Concerns:**

["NO or VERY MINOR ethics concerns only"]

**Final Justification:**

The authors performed thorough experiments both in the paper and rebuttal, and addressed most of my concerns. They also related in the post-rebuttal discussion to limitations due to the size of the input memory. These limitations should be explained clearly in the camera-ready version. I therefore raise my score to borderline accept.

**Limitations:**

The authors stated that the zero-shot performance of the VLM is required to be sufficiently good, but didn’t ablate on the matter. The authors should address limitations of frame-selection methods for longer relevant context, i.e. small portion of distractors (not just longer inputs).

**Quality:**

2

**Strengths And Weaknesses:**

Strengths:
1. The proposed method shows improved results on multiple benchmarks, outperforming existing VLMs even for short inputs that fit the context window. For inputs longer than the context window, the method prunes the input, adaptively to the question, to fit in the context window.
2. The method is simple and can essentially be applied to any VLM.
3. Evaluation is thorough and covers 4 benchmarks, 3 VLMs, and comparisons to many methods (Table 3, top left).
4. The authors show that the method dynamically adapts its focus depending on the type of question, achieving a similar portion of frames as human annotation (Figure 4).
5. Ablations on different aggregation methods are also thorough, and cover both sampling based on feature similarity using SigLIP embeddings and sampling based on VLM over captions (rather than over frames).

Major weaknesses:
1. Frame selection methods, such as the one proposed in the paper, can benefit the most in “needle-in-a-haystack” (NIAH) cases, where most of the input video is not relevant for answering the question (a large portion of “distractors”). Most long-video QA benchmarks are biased towards such cases (see [A]) and don’t reflect long-form reasoning. Frame-selection methods are therefore inherently limited, as the portion of relevant context grows, the method should adaptively select more frames, returning to the starting position where VLMs can’t truly process long inputs of relevant context. The paper does show improved results on several benchmarks with 30% and 56% relevant context frames (LVBench summarization questions and Egoschema, respectively). An evaluation on [A] with Gemini-1.5-flash should also be completed.
2. The authors review frame-selection methods that reason over extracted captions, and state that this introduces vulnerability to the captioner's mistakes. The proposed method uses the VLM alone, eliminating this vulnerability, but assumes good zero-shot performance for the frame-selection to work. In its current form, the method is applicable to models that are already state-of-the-art. I would expect an ablation on the zero-shot performance to be enough for the method to work, followed by clear requirements\benchmarks a new model should uphold in order to effectively deploy the suggested method.
3. The proposed method is simple and practical, but lacks theoretical grounding and justification other than empirical results. For example, in Table 1, partitioned context aggregation is shown to outperform hierarchical context aggregation. Assuming NIAH inputs, the coarse-to-fine strategy seems like a better idea than stratified sampling (partitioned context aggregation). Indeed, as stated by the authors, this strategy is sensitive to the coarse sampling step. This can be mitigated by performing the process multiple times and finding a consensus among runs. Otherwise, an empirical study on the distribution of the resulting context frames of the method should be provided to justify the stratified sampling choice.
4. Evaluation comparison on long videos (table 3, top right, and text) assumes that using a larger-context version of a model should result in better performance (Gemini-1.5-flash context-window to 700K tokens improves on 32K). This contradicts the premise of frame-selection, where only a small portion of frames is required, and the rest are distractors that should hurt the performance. I would appreciate the authors' opinion on the matter.

Minor weaknesses:
1. The authors should refrain from using “most relevant context” (lines 7, 9, 50, 51, Figure 2 caption) as they do not show proof of optimality.
2. Missing ablations on different context prompts.
3. Using the frame-selection justification of the model (j in Eq.4) to iteratively refine the prompt for the VLM can also be explored.

[A] CINEPILE: A Long Video Question Answering Dataset and Benchmark, Rawal et al.

---

> ### Author Rebuttal · Authors · 2025-07-30
>
> We thank the reviewer for their detailed comments and address each concern below:
>
> ---
> ### **Evaluation on Cinepile**
>
> Thank you for this suggestion. We compare our Partitioned Context Aggregation approach to baseline inference on Cinepile below. We follow the same protocol as the experiments in our paper, using a 32K / 120 frame context window for both approaches.
>
>
> Table A1: Comparison of DCA on Cinepile.
> |                    | Accuracy on Cinepile |
> |--------------------|:---------------------:|
> | Baseline inference |  53.9                    |
> | DCA (Ours)         | 55.8                    |
>
> Our DCA approach therefore still shows improvement over the baseline on this challenging dataset.
>
> ---
> ### **Is frame-selection appropriate when there is a lot of relevant context to select?**
>
> Thank you. Figure 7 in Appendix A shows that our model selects 74% of the input frames for Egoschema, and 15% of the relevant frames for LVBench. This shows that our method does indeed select the relevant context in a manner that is adaptive to the dataset. Our results on Egoschema (Figure 7 and Table 3) therefore show that when most of the input video is indeed relevant, our method is able to retain it.
>
> Finally, note that Egoschema was annotated with a "temporal certificate” (the minimum duration of the video that an annotator must look at to answer the question) of 100 seconds (or 56% of the video), and our results in Figure 7 agree with this. Similarly, Figure 6 shows that our frame selection corresponds to the "time reference” frames that are annotated for LVBench, with the most frames being selected for “summarisation” questions. Both these results emphasise the adaptivity of our approach in selecting both short and long relative-contexts from a video.
>
> We will refer to Figure 7 (in Appendix A) more clearly in the main text to make this point more clear.
>
> ---
> ### **Zero-shot performance of VLM for frame-selection**
>
> Thank you for this suggestion. Note that frame-selection for the purpose of question-answering is difficult to evaluate, as it depends on the answering model itself. From our evaluation datasets, only LVBench contains human-annotated “time-reference” frames which we use below to evaluate our frame selection.
>
> In the table below, we extend our analysis from Figure 10 of Appendix B. We evaluate the precision and recall of our selected frames, with respect to the annotated “time-reference” frames, where a “true positive” denotes a selected frame which falls within the annotated time-references for that video, and a “false positive” does not.
>
> We performed this analysis by varying the number of segments, $l$, in our Partitioned Context Aggregation approach. By increasing $l$, our recall increases, whilst precision remains fairly constant. We observe that the “time-reference” frames in LVBench are overcomplete, in that it is not necessary to select all of them to answer the question, which is why we can get accuracy improvements over the baseline with a fairly low recall. In fact, for $l \leq 12$, our recall is lower than the uniform frame sampling baseline, but the accuracy is higher due to the larger precision, showing that a subset of the time-reference frames is sufficient to answer the question. We find, however, that the accuracy correlates well with an additional metric which we define, “At Least One”, which denotes if at least one selected frame falls within the annotated “time-reference” frames.
>
>
> Table A2: Analysis of frame selection on the LVBench, using the human-annotated “time-reference” frames. “At Least One” denotes if at least one of the selected frames falls within the “time-reference” frames.
> |                       | Precision | Recall | At Least One | Average Num. Selected Frames | Accuracy |
> |-----------------------|:---------:|:------:|:------------:|:--------:|:--------:|
> | Baseline, uniform sampling              |   10.0        |  12.3      |    20.3          |    120  | 50.3      |
> | Partitioned, $l = 2$  |  20.9         |   2.7     |   41.6           |   13.4  | 50.6       |
> | Partitioned, $l = 4$  |  21.5         |  4.7      |  52.4            |   20.4  | 52.0       |
> | Partitioned, $l = 8$  |  20.0         |   8.4     |   63.4           |   34.8  | 56.9       |
> | Partitioned, $l = 12$ |  19.4         |  11.5      |   69.3           | 43.6  | 60.2        |
> | Partitioned, $l = 16$ |  18.2         |  14.3      |    71.1          |  48.5  | 61.5        |
>
>
> Note that the baseline, of uniformly selecting 120 frames from the video, has the lowest precision as expected since most of the frames in the video are irrelevant to the question. Moreover, as the videos in LVBench are an average of 68 minutes long, performing uniform sampling means that we sample at least one frame within the annotated time-references only 20.3% of the time. Using our proposed method increases this to 71.1% which explains our accuracy improvement. Finally, the fact that the “At Least One” metric is less than the overall question-answering accuracy in the first two rows, also shows that the Gemini VLM is also able to correctly answer a question without using any of the annotated time-reference frames in the dataset. This may be due to errors in the annotations of time-reference frames in LVBench, or the fact that the VLM can exploit language biases to answer the question.
>
> We will add this analysis to the appendix, and refer to it clearly from the main text.
>
>
> ---
> ### **Why not choose Hierarchical Context Aggregation?**
>
> Thank you for this suggestion. The reviewer is correct that a coarse-to-fine context aggregation approach is effective. However, we found that in terms of accuracy/computation trade-offs, it is not as effective as Partitioned Context Aggregation. As we are not able to upload figures, we include the raw data for Figure 4, Accuracy vs Computation, below, with a line added for Hierarchical Context Aggregation.
>
> In the revision, we will update Figure 4 to include this additional line, and to clarify this point.
>
>
> Table A3: Accuracy vs Computation Trade-off for different context aggregation methods. The table corresponds to Figure 4 in the main paper.
> |  Baseline inference  |  |  |  |  |  |  |
> |------------------------|:-:|:-:|:-:|:-:|:-:|:-:|
> | Total number of frames |  120 | 256 | 512 | 1024 | 2048 | 2700 |
> | Total number of tokens (K) | 30  | 66   | 132 | 264 | 528 | 696 |
> | Accuracy               | 50.3  | 53.6  | 56.3 | 58.1 | 58.9 | 58.9 |
>
> |  Hierarchical Context Aggregation  |  |  |  |  |
> |------------------------|:-:|:-:|:-:|:-:|
> | Total number of frames |  325 | 1104 | 1273 | 1816 |
> | Total number of tokens (K) | 83 | 284 | 328 | 468 |
> | Accuracy               | 53.1 | 58.6 | 60.3| 61.5 |
>
> |  Partitioned Context Aggregation  |  |  |  |  |  |  |
> |------------------------|:-:|:-:|:-:|:-:|:-:|:-:|
> | Total number of frames |  197 | 332 | 602 | 866 | 1515 | 2605 |
> | Total number of tokens (K) | 50  | 85   | 155 | 223 | 390 | 672 |
> | Accuracy               | 50.2  | 52.7  | 56.5 | 60.1 | 61.1 | 61.7 |
>
> |  Self-Consistency CoT  |  |  |  |  |  |  |
> |------------------------|:-:|:-:|:-:|:-:|:-:|:-:|
> | Total number of frames |  360 | 720 | 1080 | 1440 | 1800 | 2160 |
> | Total number of tokens (K) | 93  | 186   | 278 | 371 | 464 | 557 |
> | Accuracy               | 49.3  | 51.1  | 51.7 | 51.4 | 51.5 | 51.3 |
>
> Note that for our context aggregation methods, the total number of frames is based on the average number of frames selected by our method on the LVBench dataset.
>
> ---
> ### **Table 3 implies that longer-context is always better**
>
> Thank you. We would like to clarify that as shown in Figure 4, the accuracy of baseline inference saturates around 1024 frames / 264K tokens. We will therefore update the corresponding row for the baseline of Table 3 to 264 tokens, and add a clarifying note that performance saturates after this point.
>
> ---
> ### **The authors should refrain from using “most relevant context” as they do not show proof of optimality.**
>
> Thank you. We have revised the paper accordingly.
>
>
> ---
> ### **Using the frame-selection justification of the model (j in Eq.4) to iteratively refine the prompt for the VLM can also be explored.**
>
> Thank you for this suggestion. We initially performed experiments with this, but did not achieve improvements beyond our current approach, which is why we did not pursue it further.
>
> ---
> ### **Ablations of prompts**
>
> We ablated different prompting strategies in Tables 4 and 5 of Appendix A, and also detailed all prompts and other implementation details in Appendix D.

---

> > ### Comment · Reviewer_eVAj · 2025-08-06
> >
> > It’s nice to see that DCA improves results for CinePile as well. Thank you for adding HCA to figure 4 and noting about the compute/accuracy budget.
> >
> > Still, your results indicate DCA cleans the input from distractors, rather than allowing for processing of long inputs. What happens if 74% (number from Egoschema) of the frames result in more frames than can fit in memory? Please consider addressing this frame-selection limitation – if all frames are relevant context, frame-selection selects returns the entire input, which doesn’t fit the model. If relevant frames are removed, frame-selection should hurt accuracy.
> >
> > The true contribution of DCA is in curating the inputs – please consider changing your contributions accordingly. DCA improves across many benchmarks, is simple to understand and practical, therefore I raise my score to borderline accept.

---

> > > ### Author Response · Authors · 2025-08-06
> > >
> > > Thank you for your constructive feedback. We agree that the primary contribution of DCA is curating the inputs, and will revise the contributions to highlight this.
> > >
> > > **About the limitations of frame selection:** It is important to note that we use the same VLM, with the same context-limit, for both context aggregation and answering. Taking the reviewer’s example, if a video has $N$ frames, which are all relevant, and the model’s context-window is $k$ with $k \ll N$: Baseline inference would need to uniformly sample $k$ frames from this video before feeding it into the model, and accuracy would be compromised. Partitioned Context Aggregation would process smaller segments of $s$ frames at a time, but would ultimately return all $N$ frames since in this example, each frame is relevant. For the final answering stage, we would again need to uniformly sample $k$ frames from the selected set to fit the context window, and the final answer would therefore be the same as baseline inference.
> > >
> > > Therefore, given the same context-window budget, DCA’s frame-selection would not degrade compared to baseline inference, even in this extreme case where all frames in the video are relevant. We will make this point clear in the revision, thank you.

---

> > > > ### Comment · Reviewer_eVAj · 2025-08-07
> > > >
> > > > Thank you for these clarifications. I have raised the score to borderline-accept.

---

### Decision · Program_Chairs · 2025-09-17

**Decision:**

Accept (poster)

**Comment:**

This paper presents a method to do keyframe selection to support long video QA.  The final reviews are mixed, with one accept, one borderline accept and one borderline reject.  The main source of contention seems to be the experiments (or lack thereof) to strongly support the idea as it is quite simple.

Having read the reviews, responses and reviewer discussion, the AC recommends the paper to be accepted.  The benefits outweigh the weaknesses and the work will make for an interesting addition to the program.  That having been said, the authors are kindly requested to incorporate their additional experiments and explanations throughout the discussion into the camera-ready.